# Fidelity of a Motivational Interviewing Intervention for Improving Return to Work for People with Musculoskeletal Disorders

**DOI:** 10.3390/ijerph181910324

**Published:** 2021-09-30

**Authors:** Ida Løchting, Roger Hagen, Christine K. Monsen, Margreth Grotle, Kjersti Storheim, Fiona Aanesen, Britt Elin Øiestad, Hedda Eik, Gunnhild Bagøien

**Affiliations:** 1Research and Communication Unit for Musculoskeletal Health (FORMI), Division of Clinical Neuroscience, Oslo University Hospital, P.O. Box 4956, 0424 Oslo, Norway; mgrotle@oslomet.no (M.G.); kjersti.storheim@ous-research.no (K.S.); 2Department of Psychology, University of Oslo, P.O. Box 1094, 0317 Oslo, Norway; roger.hagen@ntnu.no; 3Department of Psychology, Norwegian University of Science and Technology, P.O. Box 8900, 7491 Trondheim, Norway; 4Research Institute, Modum Bad, P.O. Box 33, 3370 Vikersund, Norway; 5Division of Mental Health & Addiction, Vestfold Hospital Trust, P.O. Box 2168, 3103 Tønsberg, Norway; christineost@gmail.com; 6Department of Physiotherapy, Faculty of Health Science, Oslo Metropolitan University, P.O. Box 4, 0130 Oslo, Norway; Fionaa@oslomet.no (F.A.); brielo@oslomet.no (B.E.Ø.); hgrape@oslomet.no (H.E.); 7Nidelv Community Mental Health Centre, Department of Mental Health, Trondheim University Hospital, P.O. Box 3250, 7006 Trondheim, Norway; gunnhild.bagoien@ntnu.no

**Keywords:** fidelity, motivational interviewing, musculoskeletal disorder, sick leave

## Abstract

The objective of this study was to conduct a fidelity evaluation of a motivational interviewing (MI) intervention delivered by social insurance caseworkers, in a three-arm randomized controlled trial (RCT) for improving return to work for people on sick leave with musculoskeletal disorders. The caseworkers received six days of MI training, including an intervention manual prior to the trial onset, as well as supervision throughout the trial. The caseworkers recorded 21 MI sessions at regular intervals during the trial. An independent MI analysis center scored the recordings using the MI treatment integrity code (MITI 4). In addition, three experienced MI trainers assessed the adherence to the MI intervention manual on a 1–4 Likert scale and MI competence. Total MITI 4 mean scores were at *beginning proficiency levels* for two components (global technical, mean 3.0; SD 0.6 and the reflections/questions ratio, mean 1.1; SD 0.2) and *under beginning proficiency* for two components (global relational, mean 3.2; SD 0.7 and complex question, mean 34.0; SD 21.2). The MI trainers’ assessment showed similar results. The mean adherence score for the MI sessions was 2.96 (SD 0.9). Despite delivering a thorough course and supervision package, most of the caseworkers did not reach proficiency levels of good MI competence during the study. The fidelity evaluation showed that a large amount of training, supervision and practice is needed for caseworkers to become competent MI providers. When planning to implement MI, it is important that thorough consideration is given regarding the resources and the time needed to train caseworkers to provide MI in a social insurance setting.

## 1. Introduction

Work disability and return to work (RTW) depend upon several factors including individual, workplace, healthcare, compensation systems and social factors [1,2,3]. Despite various targeted efforts to facilitate RTW, there are no conclusive results on what constitutes an effective approach [4,5].

Motivational interviewing (MI), first developed by Miller in 1983 [6], is a widely used counseling method intended to elicit behavior change found to be effective in several randomized controlled trials (RCTs) [7,8]. The central aspects of MI are addressing ambivalence and increasing motivation for behavioral change. This method has commonly been used in the treatment of addiction [8,9], and has also proven useful for other health concerns such as weight loss [10], managing chronic illness [11] and increasing physical activity [12]. More recently, MI has been applied to people on sick leave [13], including those with musculoskeletal disorders [14], but the evidence for its effectiveness on RTW is scarce [15].

An evaluation of fidelity is highly recommended in trial design as it has implications for a study’s reliability and internal validity. Fidelity, also referred to as adherence, integrity and quality of implementation, may be defined as the extent to which delivery of an intervention adheres to the original protocol or program model [16]. Without documentation or measurement of intervention fidelity, it is not possible to determine whether unsuccessful outcomes reflect a failure of the protocol or failure to implement the intervention as intended [17]. The Motivational Interviewing Treatment Integrity (MITI) is a behavioral coding system for evaluating MI proficiency and can be used as a treatment integrity measure in clinical trials [18]. The MITI is the most frequently used instrument for assessing MI fidelity in RCTs [18,19]. The latest version (MITI 4) yields reliable and valid indicators of MI practice [18]. However, normative or other validity data to support the values for MI competence is lacking and it is therefore recommended to use the MITI 4 in combination with other assessments of MI competence [20].

The objective of this study was to evaluate fidelity of a MI intervention in a RCT for improving RTW for people on sick leave due to a musculoskeletal disorder. 

## 2. Materials and Methods

### 2.1. The MI-NAV Study 

This fidelity evaluation is part of the MI-NAV study comprising a three-arm RCT [21]. In the RCT, participants were randomized to either MI plus usual case management provided by the Norwegian Labour and Welfare Administration (NAV), or to stratified vocational advice intervention (SVAI) plus usual case management or to usual case management alone provided by NAV. This study evaluates the fidelity of the MI intervention. The fidelity of the SVAI intervention will be reported elsewhere. Inclusion criteria for the RCT were people living in the south-east of Norway, aged 18–67 years with 50–100% sick leave for ≥7 weeks due to a musculoskeletal diagnosis. The primary outcome for the RCT was the number of sick leave days from randomization to six-month follow-up. Detailed information on the design and recruitment is reported in the study protocol [21]. 

### 2.2. Usual Case Management 

In Norway, employees on sick leave are entitled to full wage compensation for up to 52 weeks. The employer is responsible for payment during the first 16 days of sick leave, after which payments are covered by the national insurance scheme through NAV. The employer must initiate a follow-up plan in cooperation with the employee before the end of the fourth week of sick leave and arrange a dialogue meeting with the sick-listed worker within week seven, which other stakeholders may attend when relevant. A second dialogue meeting, including both the employer and the sick-listed worker, is arranged by caseworkers at the local NAV office within the first 26 weeks of sick leave. NAV caseworkers can also consider whether the sick-listed individual needs further follow-up and may offer advice as well as various alternatives for treatment and intervention [22]. 

### 2.3. Recruitment of the NAV Caseworkers and Training in the MI Intervention 

A total of 15 NAV caseworkers from nine NAV offices in the south-east of Norway were trained to provide the MI intervention. An invitation was sent to the NAV offices, and caseworkers who were interested in participating in the project were recruited. A total of nine caseworkers were recruited as main MI caseworkers, and another six were included as reserves who could step in when needed. 

Prior to participant recruitment, the caseworkers attended a six-day (3 + 2 + 1 day) intensive course provided by a clinical psychologist and a psychiatrist, both experienced MI trainers. The course included MI theory and roleplay to practice MI skills. It was also strongly recommended to practice MI at the workplace between the course days. A manual of the MI intervention developed for this study was provided for later reference at the sessions. Throughout the sessions there was a focus on fostering a collaborative relationship between the caseworkers and the person on sick leave. The caseworkers were expected to increase the individual’s own motivation for their RTW and the overall aim of the MI sessions was to facilitate the RTW. During the first session, an agenda for the session was set and the sick-listed worker’s readiness to change and their RTW self-efficacy were assessed. The NAV caseworkers also assessed the participant’s level of RTW readiness according to the stages of change model [23], and adjusted the intervention according to motivational stage. In the second session, the MI caseworker aimed to map the participant’s work tasks and earlier attempts at a RTW. Information on available support from NAV during the RTW process was also provided. The assessment of the participant’s readiness to change and their RTW self-efficacy from the first session was then discussed further, and their life goals and values were explored. The second session would ultimately result in an action plan for change and/or RTW, as well as a decision on whether the participant was ready to take these steps. 

According to the RCT protocol [21], two face-to-face MI sessions were to be provided to each study participant in addition to usual case management. The first session should occur as soon as possible after group allocation, and the second two weeks later. The second session could be conducted over the telephone if necessary. 

### 2.4. Supervision 

Every second month throughout the recruitment period, the caseworkers could attend supervision organized as group meetings at one of the local NAV offices. The supervision was provided by a clinical psychologist who is an experienced MI trainer. Brief individual feedback from the MI analysis center concerning the results of the MITI scorings could also be discussed over the telephone after the scoring of each audio recording. 

### 2.5. Fidelity Assessment

The evaluation was based on audio recordings and included MITI 4 scorings and an additional expert assessment of MI competence and adherence to the MI manual developed for the RCT study. In line with the protocol [21], audio recordings of approximately 10% of the recruited sample of participants were required in order to assess fidelity of the MI intervention. The procedure for the recordings was as follows: (a) Up to three audio recordings were planned for each main caseworker during the recruitment period (with some individual variation depending on total number of participants allocated to each main caseworker). (b) Each main caseworker was asked to record one or two MI sessions within the groups of participants they were allocated to; the first group consisted of participants number 2–4, the second, participants number 10–12 and the third, participants number 15–20. (c) The specific conversations to be recorded were selected by a researcher after checking if consent forms for audio recording had been signed by the participants. Generally, the caseworkers were asked to record the first of the two MI sessions offered to each participant, however, some audio recordings of the second MI session as well as series of both sessions were also performed. If the reserve caseworkers provided MI sessions to ten participants or more, their MI conversations were also recorded.

The recorded sessions were coded using the MITI 4 tool by experienced coders at a MI-analysis center (KORUS) in Bergen, Norway [24]. The coders were independent from the study. As recommended, approximately 20–25 min of the MI sessions were transcribed and used for both global ratings and behavior counts [20]. For MI session one, transcripts were made for 20–25 min from the middle of the session. For MI session two, transcripts were made of the first 20–25 min of the session.

The MITI 4 tool includes four global ratings and 10 behavior counts to assess the verbal behavior of MI practitioners. A *technical component* of cultivating change talk and softening sustain talk and a *relational component* of partnership and empathy were scored on a five-point Likert scale (1 = low, 5 = high). The ten behavioral counts included: give information, questions, simple reflection (SR), complex reflection (CR), affirm, emphasize autonomy, confront, seek collaboration, persuade with permission and persuasion. Summary scores of the technical and relational components as well as a percentage score of complex reflections (CR/SR + CR) and a reflections-to-questions ratio (total reflections/(total questions)) were performed and cut-off values for beginning proficiency and MI competence proficiency were used [20]. 

In line with recommendations [20], an additional assessment alongside the MITI 4 scorings was performed by experienced MI trainers. The MI trainers included the two trainers responsible for the MI training and the RTW-MI manual in the study, in addition to the MI trainer who was responsible for supervision throughout the trial. Two of the trainers have been a member of the Motivational Interviewing Network of Trainers (MINT) since 2009. The MI trainer assessment included general evaluations of MI competence based on the full audio-recorded sessions. A four-point rating scale of 1–4 was used, where 1 = MI inconsistent, 2 = under beginning proficiency, 3 = beginning proficiency, 4 = MI competence. The scores were meant to capture the rater’s overall judgement of three specific dimensions, including: General assessment of the caseworker’s MI competence in the session;General MI spirit of the caseworker (characterized by partnership, empathy and support of the participant’s autonomy);The degree to which the caseworker managed to engage the participant in the conversation by the establishment of alliance, collaboration and eliciting change talk from the participant.

In addition, adherence to the MI manual was scored on a five-point Likert scale ranging from 0–4 (0 = not at all, 1= to a small degree, 2 = to some degree, 3 = to a large degree and 4 = to a very large degree). The caseworker’s adherence to the main themes in the manual was scored in the first and second MI sessions (Appendix A). The scores were assessed separately by each of the experienced MI trainers and compared during a consensus meeting. If the scores differed, consensus was reached through discussion or by using the scores decided upon by the majority of the three MI trainers.

### 2.6. Analysis 

Descriptive statistics on the caseworkers’ background variables including sex, age and number of years worked with sick leave follow-up are presented. Data from the recordings were analyzed with descriptive statistics including frequencies, percentages, means and standard deviations using SPSS version 27. The analyses included total scores and scores for the separate rounds of recordings including MITI 4 component scores and the three MI competence dimensions from the MI trainer assessment. For adherence to the MI manual, the six items relating to the main themes in MI session 1 and the eleven items from MI session 2 were analyzed separately for each session as well as for a total adherence score.

## 3. Results

### 3.1. Study Setting

Of the 15 caseworkers who attended the MI course, two dropped out shortly after training and before providing MI to participants on sick leave. Eight caseworkers were the main MI deliverers for their specific NAV offices and five were reserves, all women. The age of the main caseworkers ranged from 27 to 65 years and their RTW work experience ranged from 2 to 20 years. Of the eight main MI caseworkers, four dropped out within the first year of recruitment due to lack of capacity, lack of MI experience or the COVID-19 pandemic. Due to a lack of reserves at three of the NAV offices, only one of the main caseworkers was replaced. Therefore, five NAV offices and four caseworkers provided the intervention throughout the recruitment period. 

### 3.2. MI Supervision

The MI supervision took place throughout the recruitment period as planned, with approximately 6–8 caseworkers attending each session. The majority of those taking part in the supervision were main MI caseworkers. A total of six caseworkers also received individual feedback from the MI analysis center based on the results of the MITI 4 scorings. Of the caseworkers who did not receive this feedback, one reported lack of time as a reason and one was not contacted by the analysis center after requesting feedback. 

### 3.3. Fidelity Assessment According to MITI and Expert Evaluation

Recruitment of participants lasted from April 2019 to October 2020 and during this period 170 participants were randomly allocated to the MI arm of the RCT. A total of 119 participants (70.0%) received MI and 106 (89.1%) of these participants received both MI sessions 1 and 2. The mean number of days between MI sessions 1 and 2 was 15.4 (SD 10.2). The reasons given by the caseworkers for not providing two MI sessions was that the participant had returned to work, that the participant was on vacation or that they did not want the second session. In 10 cases, we lacked documentation on the number of MI sessions or the reasons for missing sessions. The total number of participants per caseworker ranged from 1 to 27. During the COVID-19 pandemic, 22 of the MI sessions were performed by phone (14 sessions) or video (8 sessions). 

Table 1 gives an overview of the audio recordings. There were 21 recordings from 16 (13.4%) participants, of which 5 participants had recordings of both MI sessions one and two. Eight caseworkers provided recordings in round one, five in round two and three in round three (Table 1). One main caseworker did not provide any MI sessions or recordings and left the project 10 months into the recruitment period due to lack of experience. Three of the main caseworkers who dropped out during the recruitment period were close to having ten people on sick leave recruited in the study. Only one of the reserve caseworkers, the one who replaced a main caseworker early in the study, provided an audio recording. Five of the audio recordings were of video sessions. 

Table 2 shows the total MITI 4 scores and scores for each round of recordings. The table also gives overall information on the frequency (%) for each MI proficiency level of the caseworkers. The total scores were under beginning proficiency for the global relational component and for the percent complex reflections component and at beginning proficiency for the global technical component and the relationship reflections/questions component. The results showed a tendency of improvement from first to last round of recordings, except for the MITI relationship reflections/questions component. Most caseworkers did not reach MI competence levels according to MITI 4 (Table 2). 

The results from the MI assessment by the experienced MI trainers are presented in Table 3, including total scores and scores from each round of recordings by three dimensions of MI competence. For the general assessment of MI competence dimension, most caseworkers were under the beginning proficiency level. For the general MI spirit dimension as well as for the engagement dimension, most caseworkers were at the beginning proficiency level. The percentage of caseworkers who reached the MI competence proficiency level ranged from 33.3% to 38.1% for the three MI competence dimensions. The results did not indicate an improvement by round of recording (Table 3). The total mean of the adherence scores was 2.96 (SD 0.9). Most caseworkers adhered to the MI manual to a large or very large degree (Table 4). 

## 4. Discussion

This study evaluated fidelity of a MI intervention delivered as part of the MI-NAV study [21]. Generally, the MI intervention was implemented in line with the protocol. Adherence to the MI manual was high and, of the participants who received MI, most attended the two planned MI sessions within the recommended timeframe. Furthermore, the caseworkers received more MI training than reported in previous trials [25,26], including supervision throughout the recruitment period, as recommended [27]. However, the caseworkers’ MI competence scores were generally at the beginning or under beginning proficiency level. 

The results indicated that more training was needed in order to reach a good MI competence level. However, there are few studies investigating how to successfully train counselors in MI to support a RTW [28]. Reaching proficient levels of MI has been found to be difficult despite both workshop training and additional supervision [29,30]. It is also unclear what level of competency is required for MI to be effective [31]. 

RTW is a complex process influenced by health and workplace factors, and it is possible that, in order to reach MI proficiency levels in a RTW context, more training is required than in contexts supporting more specific behavior changes such as smoking cessation or alcohol use. However, a systematic review evaluating MI competence in the field of substance abuse found that most studies failed to achieve sustained practice change in MI [29]. The social insurance setting in which the intervention was implemented may also play a role. MI has traditionally been applied in clinical settings. Caseworkers are in a different position than health professionals in that they negotiate sick leave benefits and related rights, and at the same time, act as helpers and guides during sick leave and the RTW process [32]. 

Two studies have evaluated the implementation of MI in a social insurance context and have reported several challenges [28,33]. The studies found that although the MI method was experienced as useful, it was difficult to master and translate into a sickness insurance context. A lack of coworker and managerial support, time and place for practicing MI skills, competing initiatives and high workload made implementation of the MI intervention challenging [28,33]. These results illustrate how a method will not necessarily be implemented simply because training has been provided and can help explain why, although adherence to the MI manual in our study was generally high, the MI competence scores were at low levels. In addition, in our study, for some of the caseworkers there were long intervals between the study participants and some had few participants overall, meaning their experience with practicing MI was limited. Hence, recruitment may also play a role. To our knowledge, MI fidelity has only been measured in a social insurance setting in one recent study which reported MITI fidelity scores similar to those found in our study [33]. A recent mapping review evaluating MI as a method to facilitate a RTW for individuals with musculoskeletal disorders included two RCT studies [14]. None of the included studies reported data on fidelity, but one included MI adherence rates from a self-reported checklist for MI sessions, reporting generally poor levels [13]. 

Our results showed that there was an improvement in the MITI 4 scores over recording intervals. In the first recording interval, all four MITI components showed values under the threshold of beginning proficiency, except for the reflections/questions component. However, at the last recording interval, the MITI scores showed levels of beginning proficiency or MI competence. A pattern of improvement throughout the study period was also found in another study which included several assessment periods at fixed intervals over a two-and-a-half year period during which counsellors received ongoing supervision [34]. However, great variations in MI skill between counsellors were observed, as well as fluctuations in performance in counsellors over time.

It has been recommended to use the MITI 4 version in combination with other data [20] and it has been argued that the MITI criteria for proficiency is arbitrary as provisional cut points have been set along a continuum of skills [31]. Therefore, we included an additional assessment of MI competence provided by three highly experienced MI trainers. The expert evaluation was established for the same purpose as the MITI global scores, namely, to capture the rater’s global impression or overall judgment about a specific dimension. The MI competence scores showed results similar to the MITI scores, although no clear improvement over the project period was found. A higher percentage of the caseworkers with MI competence scores across the MI competence dimensions could be seen in the assessment by the MI trainers, compared to the MITI components. Some of the differences found between the two assessments may be explained by the fact that the evaluation by experienced MI trainers applied to the whole conversation, including both the caseworkers and the sick-listed person’s segments, and not just a 20 min segment used in the MITI scoring. In addition, the expert evaluation probably included an awareness of the RTW context in which this intervention was given. In MITI 4, a random 20 min segment is the recommended duration for a coding sample, and only the caseworkers’ responses are coded. The analysis in this study was performed before the outcome evaluation of the RCT and we do not know if the level of proficiency that the caseworkers showed will influence the outcomes. One systematic review evaluating practice change in MI determined that a training method resulted in sustained practice change when over 75% of participants met beginning proficiency in MI spirit at a follow-up time point [29]. In our study, over 75% of the caseworkers met beginning or MI competence proficiency in both MI spirit and engagement in the expert evaluation. This criterion was not met for the MITI 4 dimensions, although ratings for the MITI global technical component were close to the threshold for beginning MI proficiency. 

The main strength of this study was that the recommended MITI tool was used for coding and scoring the recordings and that the scoring was performed by an independent and established MI analysis lab. Another strength is that an additional expert evaluation was included. In addition, this study was performed prior to the evaluation of outcomes from the RCT and has therefore not been influenced by those results. A possible weakness is that the MI trainers’ assessment was provided by the course instructors and the MI supervisor in the study. Although performed by experienced counsellors, it is possible that this could have affected the results. Moyers et al. argue that supervisors working closely with clinicians to implement MI with fidelity as a clinical trial proceeds are likely to be biased in a manner that compromises their ability to further evaluate those same clinicians objectively [18]. However, we found that the results from the two evaluations were similar. The MI trainers were also able to incorporate the RTW context to a larger extent by including the whole conversation in the evaluation. Another limitation is that half of the main caseworkers dropped out during the study period, which left few caseworkers available for the last recording interval. Further, three of the caseworkers who dropped out were close to providing the second round of recordings. As a result, the caseworkers who were left in the study gained more MI practice, however, more recordings from all the caseworkers could have provided the findings with greater validity. Further studies should also aim to recruit men as MI counselors in this setting. 

## 5. Conclusions

The fidelity study was implemented according to protocol. The results showed that, during the study, most of the caseworkers did not reach proficiency levels indicative of good MI competence. However, improvements in MITI 4 scores were seen in the last round of recordings, indicating that considerable practice is needed to master MI. Ensuring that caseworkers receive the necessary support and the possibility to practice MI prior to and throughout the study period is recommended. When planning research implementation, it is important that thorough consideration is given to the resources and the time needed to train caseworkers to provide MI in a social insurance setting. There is uncertainty related to the proficiency levels necessary to improve outcomes in this field which should be investigated in future studies.

## Figures and Tables

**Table 1 ijerph-18-10324-t001:** Overview of total number of audio recordings, rounds of recording and type of motivational interviewing (MI) session.

	Round 1 of Recordings: Participants 2–4 ^1^	Round 2 of Recordings: Participants 10–12 ^2^	Round 3 of Recordings: Participants 15–20 ^3^
Total number of recordings	10	6	5
Caseworkers with recordings	8	5	3
First MI session recorded	8	4	3
Second MI session recorded	2 ^4^	2 ^5^	2 ^4^

^1.^ 2–7 months of recruitment; ^2.^ 6–18 months of recruitment; ^3.^ 12–15 months of recruitment; ^4.^ Series of recordings did also include recording of session 1; ^5.^ One of these recordings also included a recording of session 1.

**Table 2 ijerph-18-10324-t002:** Mean (standard deviation) of MITI 4 scores and motivational interviewing (MI) proficiency frequencies for 21 recordings across three consecutive rounds.

MITI Domain Post Training	Recording Interval and Number of Recordings				Overall MI Proficiency, Number (%)		
	Round 1 (*n* = 10)	Round 2 (*n* = 6)	Round 3 (*n* = 5)	Total (*n* = 21)	<Beginning	Beginning	MI Level
Global technical component ^1^	2.90 (0.7)	3.17 (0.3)	3.00 (0.7)	3.00 (0.6)	6 (28.6)	13 (61.9)	2 (9.5)
Global relational component ^2^	3.05 (0.9)	3.33 (0.4)	3.50 (0.5)	3.24 (0.7)	9 (42.9)	7 (33.3)	5 (23.8)
Relationship reflections/questions ^3^	1.16 (0.6)	1.08 (0.7)	1.06 (0.5)	1.11 (0.2)	10 (47.6)	9 (42.9)	2 (9.5)
Percent complex reflections ^4^	25 (19.9)	34 (18.3)	54 (15.8)	34 (21.2)	13 (61.9)	3 (14.3)	5 (23.8)

MITI 4—The Motivational Interviewing Treatment Integrity 4; ^1.^ global technical component, threshold MI competence: ≥4, threshold beginning proficiency: ≥3; ^2.^ global relational component, threshold MI competence: ≥4, threshold beginning proficiency: ≥3.5; ^3.^ relationship reflections/questions, threshold MI competence: 2, threshold beginning proficiency: 1; ^4.^ percent complex reflections, threshold MI competence: 50%, threshold beginning proficiency: 40%.

**Table 3 ijerph-18-10324-t003:** Frequency and percent of the motivational interviewing (MI) competence scores by MI trainers for 21 recordings across three consecutive rounds.

MI Competence (1–4) ^1^	Round 1 (*n* = 10)	Round 2 (*n* = 6)	Round 3 (*n* = 5)	Total (*n* = 21)
General MI competence (*n*, %)				
Under beginning proficiency (2)	3 (30.0)	4 (66.7)	2 (40.0)	9 (42.9)
Beginning proficiency (3)	3 (30.0)	0 (0.0)	1 (20.0)	4 (19.0)
MI competence (4)	4 (40.0)	2 (33.3)	2 (40.0)	8 (38.1)
General MI spirit ^2^ (*n*, %)				
Under beginning proficiency (2)	2 (20.0)	0 (0.00)	2 (40.0)	4 (19.0)
Beginning proficiency (3)	3 (30.0)	4 (66.7)	3 (60.0)	10 (47.6)
MI competence (4)	5 (50.0)	2 (33.3)	0 (0.0)	7 (33.3)
Engagement ^3^ (*n*, %)				
Under beginning proficiency (2)	2 (20.0)	1 (16.7)	1 (20.0)	4 (19.0)
Beginning proficiency (3)	4 (40.0)	4 (66.7)	2 (40.0)	10 (47.6)
MI competence (4)	4 (40.0)	1 (16.7)	2 (40.0)	7 (33.3)

^1.^ MI competence assessed by experienced MI trainers. A scale of 1–4 was provided, where 1 = MI inconsistent, 2 = under beginning proficiency, 3 = beginning proficiency and 4= MI competence; ^2.^ MI consistency relationship characterized by empathy, partnership and support of the person’s autonomy; ^3^^.^ Based on the establishment of alliance, collaboration and change talk.

**Table 4 ijerph-18-10324-t004:** The caseworkers’ adherence to the Motivational Interviewing (MI) manual’s main themes.

	MI Session 1 ^1^ (*n* = 15)	MI Session 2 ^2^ (*n* = 6)	Total (*n* = 21)
Mean (SD)	2.97 (0.9)	2.94 (0.8)	2.96 (0.9)
Median (range)	3.00 (0.0–4.0)	3.0 (2.0–4.0)	3.0 (0.0–4.0)
Frequency (%)			
≤2	3 (20.0)	1 (16.7)	4 (19.1)
3	5 (33.3)	0 (0)	5 (23.8)
4	7 (46.7)	5 (83.3)	12 (57.1)

^1.^ MI session 1 adherence score consisted of 6 items; ^2.^ MI session 2 adherence score consisted of 11 items; items were scored on a five-point rating scale: 0 = not at all, 1 = to a small degree, 2 = to some degree, 3 = to a large degree and 4 = to a very large degree. Three of the recordings had missing data for 2–4 items.

## Data Availability

The datasets used and/or analyzed during the current study are available from the corresponding author on reasonable request.

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
