# Peer review of "Fidelity of a Motivational Interviewing Intervention for Improving Return to Work for People with Musculoskeletal Disorders"

_ijerph, 2021, doi:10.3390/ijerph181910324_

Round 1

Reviewer 1 Report

General comments

I would like to thank the authors for this interesting manuscript titled “Fidelity of a Motivational Interviewing intervention for improving return to work for people with musculoskeletal disorders”. However, I also did notice minor errors throughout the paper that require editing.

1.Introduction

Line 49: I recommend you delete reference 16, it already appears on line 51 of the same paragraph.

  1. Materials and Methods

Line 79, 86…: I recommend you standardize how to write the numbers, for example, in lines 79 and 86 they are in "number" and in lines 89, 92, 126, 147 ... they are written in "letter".

Line 96: there is double space behind dot, you should change to single space.

Line 173: I don´t understand the reference to table A1, will it be table 1???.

  1. Results

Nothing to add in this section

  1. Discussion

Line 322: there is double space behind “and”, you should change to single space.

Line 354: I recommend you change the reference number 18 to line 357, after the word "objectively".

  1. Conclusions

Nothing to add in this section

I would like to suggest to the authors to correct the minor errors in the paper in order to resubmit it.

I wish the authors the best of luck in their editing process.

Reviewer 2 Report

It's a nice manuscript.

Would you please delete lines 187-189?

Please correct the format of tables on page 6, e.g. from line 248, move 4 to the next line, correct table 2 column 1 reflection/questions, keep line 263-264 with Table 3.

Please crosscheck line 230.

Chapter 4 should be divided by subheadings, or the only subheading is eliminated.

Reviewer 3 Report

I really appreciate the opportunity to review this manuscript entitled “Fidelity of a Motivational Interviewing intervention for improving return to work for people with musculoskeletal disorders”. This is important to assess the importance of a Motivational Interviewing for people with musculoskeletal disorders.  I only remark some issues (most of them in methods) in order to improve the quality of this manuscript.

The abstract is clear but it is important to explain the design, it there any kind of control group? Also in the Keywords, is MITI a recognized keywords in the thesaurus?  

Introduction was well structure and shows the necessity for this research. The aim of the paper is easy to understand. Authors talked how to do the evaluation (line 62-65). This aspect should be moved to methods.

At the methods section, there are some questions that should be review. If this study, is a part of a big research project (line 68) it should be explained.

About the inclusion criteria, why authors chose patients age of 18–67 years with 50-100% sick leave for ≥ 7 weeks due to a musculoskeletal diagnosis.? It is a really big percentage change from 50 till 100. About the experimental procedures. In the fidelity assessment, why approximately 10% of the recruited sample were to be made in order to assess fidelity?

Results were clear. Discussion summarize and explain in a good way the finding but, from my point of view it would be interesting to discuss about gender differences (all participants are women).  Also, the caseworkers’ MI competence scores were generally at the beginning or under beginning proficiency level, which can be future lines of research? And how to avoid half of the main caseworkers dropped out during the study period?

Conclusions were correct.
